# Molecular Mechanisms of Autophagy in Cancer Development, Progression, and Therapy

**DOI:** 10.3390/biomedicines10071596

**Published:** 2022-07-05

**Authors:** Veronica Angela Maria Vitto, Silvia Bianchin, Alicia Ann Zolondick, Giulia Pellielo, Alessandro Rimessi, Diego Chianese, Haining Yang, Michele Carbone, Paolo Pinton, Carlotta Giorgi, Simone Patergnani

**Affiliations:** 1Laboratory for Technologies of Advanced Therapies (LTTA), Department of Medical Science, University of Ferrara, 44121 Ferrara, Italy; veronicaangelamaria.vitto@unife.it (V.A.M.V.); silvia.bianchin@unife.it (S.B.); giulia.pellielo@unife.it (G.P.); alessandro.rimessi@unife.it (A.R.); diego.chianese@unife.it (D.C.); paolo.pinton@unife.it (P.P.); 2Thoracic Oncology, University of Hawaii Cancer Center, Honolulu, HI 96816, USA; aliciaz@hawaii.edu (A.A.Z.); haining@hawaii.edu (H.Y.); mcarbone@cc.hawaii.edu (M.C.); 3Department of Molecular Biosciences and Bioengineering, University of Hawai’i at Manoa, Honolulu, HI 96816, USA

**Keywords:** autophagy, cancer, tumor suppression, tumor promotion, anoikis, therapy

## Abstract

Autophagy is an evolutionarily conserved and tightly regulated process that plays an important role in maintaining cellular homeostasis. It involves regulation of various genes that function to degrade unnecessary or dysfunctional cellular components, and to recycle metabolic substrates. Autophagy is modulated by many factors, such as nutritional status, energy level, hypoxic conditions, endoplasmic reticulum stress, hormonal stimulation and drugs, and these factors can regulate autophagy both upstream and downstream of the pathway. In cancer, autophagy acts as a double-edged sword depending on the tissue type and stage of tumorigenesis. On the one hand, autophagy promotes tumor progression in advanced stages by stimulating tumor growth. On the other hand, autophagy inhibits tumor development in the early stages by enhancing its tumor suppressor activity. Moreover, autophagy drives resistance to anticancer therapy, even though in some tumor types, its activation induces lethal effects on cancer cells. In this review, we summarize the biological mechanisms of autophagy and its dual role in cancer. In addition, we report the current understanding of autophagy in some cancer types with markedly high incidence and/or lethality, and the existing therapeutic strategies targeting autophagy for the treatment of cancer.

## 1. Introduction

The term “autophagy” was first coined fifty years ago by Christian de Duve in 1963 [1,2]. There are three distinct types of autophagy: macroautophagy, microautophagy and chaperone-mediated autophagy (CMA) [3], and macroautophagy is often referred to as autophagy [4]. Autophagy is one of the major cellular and organellar quality-control systems, and is evolutionarily conserved from yeast to humans, as it is fundamental to maintain physiological homeostasis in response to environmental and cellular stresses [5]. Several different forms of stress may induce this highly catabolic mechanism, such as danger-associated molecular patterns (DAMPs), mitochondrial damage, reduced cellular energy, redox stress, oxidative stress, hypoxia, endoplasmic reticulum (ER) stress, anoikis due to cellular detachment from the extracellular matrix (ECM), pathogenic infections, nutrient deficiency (the reduction of glucose and amino acids) and the depletion of growth factors [6,7,8]. To preserve intracellular homeostasis, these stress signals stimulate autophagy to help clear damaged or aged organelles, degrade misfolded or denatured soluble proteins into amino acids in the cytoplasm, and enhance biosynthesis and energy production. In these ways, autophagy is a self-protective mechanism to promote cell survival [9,10]. Excessive or insufficient autophagy caused by chronic stress may lead to the accumulation of toxic substances and, consequentially, the onset of many pathological conditions, such as lung disease, diabetes, obesity, neurodegeneration, and especially cancer. Insufficient autophagy due to severe stress may cause self-degradation and cell death [11,12,13,14]. In cancer, autophagy acts as a tumor suppressor by preventing the accumulation of damaged organelles and proteins, and enhances tumor growth by promoting cell survival [15]. The majority of studies demonstrate that autophagy can be overactivated, dysregulated or suppressed in cancer cells, and its roles in regulating cancer are dependent on the different stages of tumorigenesis. Moreover, recent reports show that autophagy is causally linked to some of the hallmarks of cancer (i.e., excessive cell proliferation, growth suppressor evasion, cell death resistance, anoikis resistance, cellular immortalization, angiogenesis, and invasion and metastasis) [16,17]. However, other studies report that during tumor initiation, autophagy suppresses tumor formation by protecting cells from DNA damage and preventing cell transformation. Autophagy has been shown to increase cancer cell mortality. Conversely, autophagy promotes cancer cell survival in unfavorable conditions, such as hypoxia or nutrient deficiency. In addition, autophagy stimulates tumor cell metastases by increasing invasive properties, enhancing vascularization and enabling apoptotic evasion following treatment with chemotherapeutic agents and radiation [18]. While careful consideration of a patient’s genetic background, cancer type and stage of tumorigenesis is essential, therapeutic approaches to modulate autophagy in cancer have the potential to improve anticancer treatments [19].

## 2. Biological Mechanisms of Autophagy

Autophagy activation consists of several closely related sequential steps. In these distinct stages, there are more than 40 autophagy-related (ATG) proteins involved, of which only 19 proteins are required for autophagosome formation [20]. In this section, we briefly describe the stages of autophagosome formation and introduce the major proteins and complexes involved in the biological mechanisms of autophagy.

The stages of autophagy are: initiation, nucleation, elongation and maturation, fusion, degradation and recirculation of intra-autophagosomal contents in the autophagolyosome [5,21].

The initiation step of autophagy is triggered by nutrient fluctuations, intracellular or extracellular stimuli and different phosphorylation/dephosphorylation events. From a molecular perspective, the autophagy activation is mainly orchestrated by nutrient-responsive kinases that have opposite activity [22]. These kinases, 5′ AMP-activated protein kinase (AMPK) and mechanistic (or mammalian) target of rapamycin (mTOR), are major positive and negative regulators of autophagy, respectively [23].

In the cell, mTOR forms two structurally and functionally distinct complexes, mTORC1 and mTORC2. mTORC1 regulates cellular metabolism and responds to external and internal signals by sensing the presence of growth factors and available nutrients to regulate autophagy [24]. mTOR1 is comprised of three main proteins: mTORC, MLST8 and Raptor [25]. In particular, the presence of certain amino acids such as leucine, glutamine and arginine allows mTORC1 to maintain autophagy inhibition [26,27]. In contrast with mTOR, AMPK is a serine/threonine kinase comprised of a catalytic α subunit, and regulator β and γ subunits, which are involved in various pathological processes and in the regulation of different cellular pathways [28]. This kinase senses intracellular energy levels and maintains intracellular calcium (Ca^2+^) homeostasis in response to glucose starvation or decreases in ATP/AMP ratio caused by energy stress (Figure 1) [29]. AMPK is also regulated by direct allosteric activation of AMP at the γ subunit that results in phosphorylation of threonine-172 (Thr172) of the catalytic α subunits [30]. On the one hand, AMPK can stimulate ATP production (i.e., glucose metabolism and uptake) through the activation of many transcription factors involved in catabolism pathways. On the other hand, it can reduce ATP utilization by inhibiting the activity of numerous transcription factors involved in anabolic routes such as carbohydrate, protein and lipid biosynthesis [31].

The beginning of autophagy initiation is mediated by a multi-protein complex known as Unc-51-like kinase 1 (ULK1)-complex, consisting of FIP200, ULK1, ATG13 and ATG101. Under nutrient-rich conditions, mTOR negatively regulates the initiation of autophagy by hyper-phosphorylating ATG13 and blocking the interaction of ATG13 with ULK1 and FIP200 (Figure 1) [32]. In this complex, the serine/threonine protein kinase ULK1 is the essential player necessary for autophagy initiation; thus, ULK1 is regulated by phosphorylation events at 30 different sites [33].

Upon nutrient starvation, ULK1 can be activated directly and indirectly by AMPK. ULK1 is directly regulated by AMPK through phosphorylation on the three serine residues, which leads to ULK1 activation, and as a result, mTOR dissociates from the ULK 1 complex [34]. ULK1 is indirectly regulated by the downregulation of mTOR activity and the dissociation of mTOR from the ULK1 complex. Subsequently, activated ULK1 can phosphorylate itself as well as its binding partners ATG13 and FIP200, and the phosphorylated ULK1 complex functions to control the initiation stage of autophagy and form the phagophore. ATG13 is a crucial protein that facilitates the binding of ULK1 to a pre-autophagosomal structure (PAS, also termed the Phagophore Assembly Site), an essential site of the autophagosome formation, and cytoplasm to vacuole targeting (Cvt). In addition, ATG13 enables the interaction of FIP200 with ULK1. FIP200 acts as a scaffold that modulates the anchoring of all autophagy-related (ATG) proteins onto the PAS [35,36]. Once all the ATG proteins are localized and anchored to the PAS, the autophagy process initiates [35,37]. Consequently, the activation of the ULK1 complex stimulates the nucleation of the phagophore through the recruitment and activation of the class III phosphatidylinositol 3 kinase (PI3KC3) complex. The PI3KC3 is essential for initiation and maturation of the phagosome. The PI3KC3 complex core is comprised of class III PI3K, vacuolar protein sorting 34 (VPS34), vacuolar protein sorting 15 (VPS15), p150, Beclin-1 and ATG14, and these components are critical for PI3K activity and autophagy induction [38,39,40]. Among them, Beclin-1 is the core of PI3KC3 complex and is regulated by numerous negative and positive regulators that interact with Beclin-1 and influence the autophagy pathway. The positive regulators of the PI3K complex are: UV resistance-associated gene (UVRAG), which competes with ATG14 for binding to Beclin-1, and BAX-interacting factor 1 (Bif-1) which interacts with Beclin-1 via UVRAG to promote PI3K complex activity and supports membrane curvature [38]. Moreover, Rubicon and antiapoptotic family members (such as Bcl-2, Bcl-XL and Mcl-1) are fundamental negative regulators of autophagy through their interaction with Beclin-1 (Figure 1) [41]. As a result, activation of the PI3KC3 complex generates local increased phosphatidylinositol-3-phosphate (PI3P) at the level of the ER. Importantly, PI3KC3 activation promotes formation of omegasomes, which are necessary for the nucleation/autophagosome formation, and stimulates the recruitment of effectors, such as WD-repeat protein interacting with phosphoinositides (WIPI) family proteins [42] and the double FYVE domain-containing protein 1 (DFCP1) [43,44]. Subsequently, the autophagosome elongation is mediated by the ATG9 transmembrane protein, which regulates the trafficking of cellular components from mitochondria, ER (endoplasmic reticulum), plasma membrane, endosome and the Golgi complex [45,46,47]. There are two ubiquitin-like conjugation systems (i) ATG12–ATG5 and (ii) ATG8 (or LC3) that participate in autophagosome closure, maturation and the recruitment of additional autophagy machinery [48]. The ATG12–ATG5 complex involves the conjugation of the ATG12 protein to ATG5 through action of the E1-like enzyme ATG7, and the E2-like enzyme ATG10 before conjugation with ATG5. Consequently, the newly formed complex ATG5-ATG12 binds the ATG16L1 protein to form a larger multimeric complex ATG5-ATG12-ATG16 with E3 ligase activity [49]. The ATG8/LC3 pathway begins with the proteolytic cleavage of the C-terminus—LC3 by the ATG4B protease, which induces the soluble form LC3-I, through the cleavage of the C-terminus—LC3. This form of LC3 conjugates lipids that are generally phosphatidylethanolamines (PE) via ATG7 (E1-like)—ATG3 (E2-like) proteins and the ATG5-ATG12-ATG16 complex (E3-like activity). This complex progresses into an autophagy membrane-bound LC3-II form to ensure expansion and elongation of autophagy membranes and their closure (Figure 1) [6]. Finally, LC3-II is delipidated and localized to the outer autophagosomal membrane. The lysosomal membranes form autolysosomes by fusing with the outer membrane of mature autophagosomes [50]. In this step, different complexes are involved, including integral lysosomal proteins (e.g., LAMP2), RAB proteins (e.g., RAB5 and RAB7) and SNARE complexes (e.g., VAMP8, STX17). These complexes degrade the contents of the autophagosome via several hydrolytic enzymes, including cathepsins [6]. The cellular components that are shuttled by autophagosomes are broken down into their building blocks (e.g., proteins into amino acids), and only after this breakdown can the contents be transported from the lysosomal lumen into the cytosol.

## 3. Transcriptional Regulation of Autophagy

In 1999 Kirisako et al. have observed that the induction of autophagy after nitrogen starvation in yeast results in the transcriptional upregulation of Apg8/Aut7p autophagy-related gene [51]. However, in 2011, Settembre and its collaborators discovered the master regulator of lysosomal pathways that controls the several autophagy-related genes, called transcription factor EB (TFEB) [52]. Nowadays, it has been demonstrated that TFEB can act on different steps of the autophagy process regulating the expression of genes for: (i) autophagy initiation such as Beclin-1, WIPII1, ATG9B and NRBF2; (ii) autophagosome elongation (ATG5); (iii) autophagosome fusion and trafficking with lysosomes (UVRAG, RAB7) [52,53]. Moreover, Ca^2+^- and calmodulin-dependent serine/threonine phosphatase calcineurin may induce the translocation of TFEB into the nucleus, thus regulating the autophagic process through the expression of the autophagy genes [54].

TFEB is a member of the microphthalmia family of bHLH-LZ transcription factors (MiT-TFE) that includes TFEC, TFE3 and MITF [55]. Several studies demonstrated that these transcription factors also regulate the autophagosomal and lysosomal biogenesis [52,56,57]. Interestingly, it has been shown that the activity and the localization of MiT/TFE family is regulated by mTORC1. In particular, mTORC1 induces their inactive form, binding and enhancing the phosphorylation, sequestering them in the cytosol and preventing their translocation into the nucleus, while in their active dephosphorylated form, they translocate into the nucleus, stimulating autophagy and lysosomal biogenesis through the increase in autophagosomal and lysosomal gene expression [58,59,60,61].

Similar to the MiT/TFE family, also another class of transcription factors called forkhead box O (FOXO) are activated in response to external energy changes, growth factors stimulation and nutritional status to regulate the expression of genes involved in autophagy regulation [62]. FOXO family includes four members: FOXO1, FOXO3, FOXO4 and FOXO6. FOXOs can modulate autophagy in either transcription-dependent or transcription-independent fashion [63,64]. Among the different FOXOs members, FOXO3 was the first member of the family identified as a transcriptional regulator of several autophagy genes (e.g., ATG4, ATG12, Beclin-1, LC3, ULK1/2 and VPS34) [64,65]. The transcriptional activity of FOXO3 is regulated by its translocation from the cytosol to the nucleus and vice versa. Indeed, in the presence of growth factors, FOXO3 is phosphorylated by Akt, which causes its cytoplasmic retention and consequent inhibition of its transcriptional activity [66,67,68]. Interestingly, it has been demonstrated that FOXOs not only regulate the transcription of autophagy genes, but they can induce autophagy in a transcriptional-independent way. For example, through the dissociation from sirtuin-2 (SIRT2), FOXO1 directly interacts with key autophagy regulators (such as ATG7), favoring the induction of autophagy [64]. Another member of FOXOs involved in the modulation of autophagy is FOXO3.

Recent studies suggested that the nuclear receptor farnesoid X receptor (FXR) strongly suppresses autophagy in the liver during feeding conditions of mice [69,70]. FXR controls postprandial metabolic responses, and it is activated by elevated bile acid levels, an event correlated to repression of autophagy [71,72]. Indeed, it has been demonstrated that FXR activation caused by high levels of bile acids inhibits the transcriptional activity of the fasting-activated cAMP response element-binding protein (CREB). This happens through its interaction with CREB, destroying the complex formed by CREB and its coactivator CRTC2, resulting in formation of a repressive transcriptional complex and inhibition of autophagy genes. Consistently, during nutrient-deprived fasting, FXR is inhibited, while CREB/CRTC2 complex can activate transcription of autophagy-related genes including ATG7, ULK1 and TFEB that are involved in multiple autophagic processes [70]. Lee et al. have demonstrated that nutrient availability inhibits FXR, while it activates nuclear receptor Peroxisome Proliferator Activated Receptor alpha (PPARα), which acts as a transcriptional activator. These two nuclear receptors, PPARα and FXR, compete for binding to shared sites in autophagy gene promoters, with opposite transcriptional outputs [70]. Interestingly, during starvation, TFEB is activated through an autoregulatory feedback loop and exerts a transcriptional control on lipid catabolism via PPARα and its coactivator peroxisome proliferator activated receptor gamma 1 alpha (PGC1α). This, therefore, suggests a connection between the autophagy pathway and cellular energy metabolism through a transcriptional mechanism [73]. All this evidence emphasizes that both of these circuits cooperate in the coordination of autophagy with other metabolic processes.

Moreover, the transcription factor nuclear factor-κB (NF-κB) and the cell cycle regulator E2F-1, through the regulation of BNIP3 expression, can regulate autophagy [74,75]. BNIP3 is a protein with homology to Bcl-2 in its BH3 domain and is a direct transcriptional target of hypoxia-inducible factor 1 (HIF1) [76]. Indeed, the expression of BNIP3 is transcriptionally induced by ischemic or hypoxic stress, and its promoter contains DNA-binding elements for NF-κB and E2F-1 that are absent from the other Bcl-2 death factors. These unique properties of BNIP3 highlight its importance in its role as a potent inducer of autophagy [77,78,79,80]. During hypoxia, E2F1 binds the promoter of BNIP3 to induce its expression and activate autophagy. However, the expression of other autophagy genes such as ULK1, LC3 and ATG5 can be promoted by this transcription factor [81]. Normoxia reduces the occupancy of E2F1 on the BNIP3 promoter, thus allowing NF-kB to constitutively bind to the promoter of BNIP3 repressing its expression [82].

Notably, several studies demonstrated that p53, a tumor suppressor that triggers apoptosis via multiple pathways, including cell cycle arrest, can modulate autophagy based on its cellular location. Nuclear p53 can transactivate genes that promote autophagy through transcriptional induction of the damaged-regulated modulator (DRAM), a lysosomal protein which induces autophagy through a yet not identified mechanism, and Sestrins protein (Sestrin 1 and 2), which activates AMPK, while inhibiting mTORC1 lysosomal recruitment [83,84,85]. Subsequently, Kemzelmann et al. have shown on mouse embryonic fibroblasts (MEFs) upon DNA-damage that p53 controls the expression of specific genes for autophagy through experiments based on CHIP-SEQ and RNA-SEQ analysis. The genes involved in the regulation by p53 are: (i) for autophagy initiation LKB1, ULK1 and ULK2; (ii) autophagosome maturation ATG4, ATG7 and ATG10 [86]. In addition to this, upon DNA damage, p53 can also regulate the expression and activity of FOXO3 as well as the nuclear translocation of TFEB/TFE3 [87,88,89,90]. Meanwhile, cytoplasmic p53 inhibits AMPK and activates mTOR, thereby acting as a negative regulator of autophagy [91,92]. Although p53 is one of the most studied tumor suppressors, unfortunately, its role in the regulation of autophagy pathway is not yet fully clear.

## 4. Autophagy and Anoikis

Interaction between cells and extracellular matrices (ECM) requires complex cellular cross-talk through focal adhesions (FAs). FA are plasma membrane-associated complexes that physically connect the cell cytoskeleton to the ECM through cell adhesion proteins referred to as integrins. Integrins deliver the exchange of extracellular signals and intracellular signals from the cell membrane to the inside of the cell and vice versa, by interacting with signaling molecules and cytoskeletal proteins [93]. Integrins interact with and recruit the components of the ECM, such as: collagen, fibronectin, vitronectin, laminin, talin, vinculin, paxillin and zyxin [94]. FA complexes are regulated by intracellular (nonreceptor) tyrosine kinases called focal adhesion kinase (FAK) and Src [94]. It has been observed that these complexes are often altered in metastatic cancer cells. The maintenance of interactions between neighboring cells and cells with the ECM is crucial for parenchymal cell survival and for sustaining proper cellular function and morphology. When cells lose the attachment to the ECM, anoikis, detachment-induced apoptosis, is activated [95,96]. Anoikis is a cellular self-defense strategy that has been implicated as a physiological barrier to metastasis [96]. This process occurs following cellular detachment from the ECM through activation of intrinsic signaling pathways promoting the assembly of BAX-BAK oligomers on the outer mitochondrial membrane mediated by the stimulation of pro-apoptotic BH3-only proteins, BIM and BID [96]. Additionally, the loss of adhesion activates the extrinsic signaling pathways, increasing expression of Fas and FasL and the downregulation of FLIP, which induces anoikis [96]. However, some reports show that the dissociation of FA from ECM in normal tissues induces antiapoptotic signals by activating autophagy as a failsafe mechanism to allow cells to survive only if they are able to reestablish attachment to the ECM, thus delaying the onset of anoikis [97]. Conversely, cells that underwent malignant transformation may utilize autophagy to evade anoikis and promote metastasis and cell motility through blood or lymphatic circulation to a distant tissue [98]. Moreover, cell detachment from ECM caused by integrins degradation suppresses survival signals such as NF-kB, PTEN and ERK due to the activation of FAK-Src [96]. ROS also play an important role in cell survival during anoikis. ROS production following cellular detachment from the ECM leads to the activation of Src, EGFR/Erk1/2 and autophagy and enables anoikis resistance [99]. Fung C. et al. demonstrated that autophagy promotes epithelial cell survival during anoikis through the stable reduction of ATG5 or ATG7 in MCF-10A [100]. In addition, RNA interference-mediated depletion of autophagy regulators (ATGs) inhibits detachment-induced autophagy, reduces clonogenic recovery after anoikis and promotes apoptosis by activating caspase-3 [100]. Furthermore, the detachment of hepatocellular carcinoma cells from the ECM allows anoikis escape through autophagy activation driven by destabilization of the mTORC1 complex and upregulation of BNIP3 (BCL2 Interacting Protein 3) via the ERK/HIF-1α pathway [101]. Autophagy can be also negatively regulated following detachment through expression of the non-coding micro RNAs (miRNAs) that promote anoikis. Moreover, miR-181 increased the metastatic potential of breast cancer cells by promoting EMT, anoikis resistance, migration and invasion. Additionally, autophagy inactivation increased the expression of the proapoptotic molecule Bim, which sensitized metastatic cells to anoikis [102]. Zhu et al. also demonstrated that the miR-30a can negatively regulate Beclin-1 expression, which led to autophagy suppression and increased cell death caused by anoikis [102]. Functional studies confirmed that miR-30a prevents anoikis inhibition in some types of tumors, both in vivo and in vitro [103,104]. Moreover, in human cancer cell lines with high levels of PI3K or oncogenic RAS, autophagy inhibition induced resistance to anoikis and reduced proliferation in anchorage-independent conditions [105,106]. Taken together, these results indicate that autophagy activation due to the loss of integrin-mediated ECM contact promotes anoikis resistance and adaptation in both normal and transformed cells. However, the regulatory mechanisms of the delicate balance between anoikis and autophagy are still poorly understood.

## 5. Autophagy and Apoptosis

Apoptosis is a caspase-mediated programmed cell death and is triggered by different mitochondrial stimuli or by molecules binding to the cell-membrane receptor. This type of death is characterized by cell shrinkage, cell membrane blebbing, chromatin condensation and nuclear and chromosomal DNA fragmentation [107,108,109,110]. Apoptotic cell death includes two pathways: intrinsic and extrinsic [111]. The intrinsic pathway is activated by various intracellular stimuli (such as growth factor deprivation, oxidative stress and DNA damage) and is based on the formation of a complex named apoptosome composed of apoptotic protease-activating factor, procaspase-9 and cytochrome c. However, the release of cytochrome c is controlled by several members of the Bcl-2 family, such as Bax, Bak, Bcl-2 and Bcl-xL, through the regulation of the mitochondrial membrane permeabilization. Meanwhile, the extrinsic pathway is induced by the binding of death ligands to death receptors of the TNF receptor superfamily. This interaction induces the assembly of the death-inducing signaling complex (DISC), composed of the Fas-associated death domain (FADD) protein, pro-caspase-8 and pro-caspase-10. Consequently, DISC activates the effector caspases (caspase-3, 6 and 7) to cleave the Bcl-2 family member Bid into tBid to activate the mitochondria-mediated intrinsic apoptotic pathway or directly induce cell death [111].

Similar to autophagy, apoptosis is an essential pathway for both degradation and maintenance of cellular homeostasis. It is not surprising that although they are characterize by different mechanisms of activation and a proper molecular authenticity, several proteins cooperate to control both apoptosis and autophagy. Here, we concentrate on some molecules that have several connections between these two pathways.

In particular, Beclin-1 is a key molecule involved in the autophagosome formation that cross-regulates autophagy and apoptosis through direct interaction with B-cell lymphoma 2 (Bcl-2) family proteins, which inhibit cytochrome c releasing from mitochondria [112,113]. When the antiapoptotic protein Bcl-2 binds to Beclin-1 through the BH3 domains, it inhibits the process of autophagy by blocking the activity of Beclin-1 and the occurrence of endogenous apoptosis. Consistently, it has been shown that an overexpression of Bcl-2 family proteins can inhibit autophagy [114,115,116].

On the other hand, NOXA and other BH3-only family proteins can promote autophagic cell death by deregulating the Bcl-2 family members that bind Beclin-1 [117]. In addition, Beclin-1 may also be the target of several endoproteases that hydrolyze peptide such as caspase-3 and caspase-8. The result of the cleavage of Beclin-1 produces a truncated protein that is unable to promote autophagy, thus modifying the cell’s fate from autophagy to apoptosis [118].

At the same time, caspases have been shown to inhibit autophagy through a mechanism of the cleavage of autophagy-related proteins [119,120]. Indeed, it has been shown that the fragment product of ATG5 cleavage is unable to promote autophagy due to its switch from cytosol to mitochondria, and binds to the antiapoptotic protein Bcl-XL, resulting in mitochondrial dysregulation and subsequent induction of cyt c release. This suggests an essential role of caspases in the inhibition of autophagy [121].

Recent studies demonstrated that other ATGs proteins may also be involved in the regulation of the apoptotic pathway. In particular, the ATG12–ATG3 complex has an autophagy-independent function that can regulate apoptosis, although they are both essential components of the autophagy ubiquitin-like conjugation machinery required for autophagosome formation [122]. Furthermore, ATG3 mutations prevent conjugation to ATG12 and, through the mitochondrial pathway, inhibit apoptosis in an autophagy-independent means, thereby highlighting that the effect on cell death is associated with altered fragmentation of mitochondria. All this evidence shows that although components of the autophagy machinery can control the apoptosis pathway, they may do so in a mode independent of autophagy. Because of the potential clinical impact covered by the modulation of these cellular pathways, more efforts should be made by researchers to improve the knowledge about the molecular mechanisms which link these important cellular pathways.

## 6. Autophagy and Ferroptosis

Iron is an essential element of the physiological function of the human body. It regulates a plethora of cellular processes, such as DNA replication and nucleic acid repair, regulation of energy metabolism and functioning of the mitochondrial electron transport chain. At the same time, excess iron accumulation is deleterious and toxic for the human body, because it promotes the formation of hydroxyl radicals which can cause peroxidation of lipids, proteins, carbohydrates and nucleic acid. Furthermore, the presence of augmented levels of hydroxyl radical can seriously damage the cell and provoke cell death. To avoid iron overload and subsequent cellular damage, cells present a fine-controlled mechanism composed of specific iron-capturing, -storage and -transport molecules (such as iron-responsive element binding proteins, transferritin and ferritin) [123], which cooperate to create a cytosolic iron pool that can be utilized by metabolic pathways or be released from the cells by specific iron exporters named ferroportin 1 [124].

Interestingly, it has been demonstrated that alteration in iron homeostasis and in regulators of iron trafficking and storage provoked a form of regulated cell death (ferroptosis) characterized by iron accumulations, which induce iron-dependent lipid peroxidation, depletion of glutathione depletion and consequent inactivation of glutathione peroxidase-4 [125]. Ferroptosis is involved in the onset and development of several human diseases, particularly neurodegeneration, cardiovascular diseases and diverse cancer types and, recently, it has been demonstrated that ferroptosis may be regulated by a particular form of autophagy named ferritinophagy. During ferritinophagy, ferritin results as degraded, thereby causing iron overload and consequent increases in oxidative stress that cause lipid peroxidation, damage to cellular membranes and finally, cell death. To regulate ferritinophagy, a specific mechanism exists in which nuclear receptor coactivator 4 (NCOA4) acts as the cargo receptor, since it interacts and transports ferritin toward autophagosomal membranes for its degradation [126,127]. Consistently, over-expression of NCOA4 increases the intracellular levels of iron and the sensitivity to ferroptosis, while NCOA4 down-regulation reflects in lower iron levels and activation in ferroptotic cell death [128]. However, the exact mechanism by which NCOA4 brings ferritin to the lysosomal compartment remains unclear. Despite the first investigation accounting a canonical ATG8-dependent autophagy delivery of NCOA4-ferritin to lysosome, further investigations also unveiled an alternative transport mechanism involving VPS34, TAXBP1, ATG9A and FIP200 [129].

Considering its role in regulating iron availability, it is not surprising that ferritinophagy regulates erythropoiesis [130]. As confirmation, NCOA4 knockout mice display disrupted iron homeostasis which leads to a decrease in serum iron and anemia [130]. Ferritinophagy is also involved during the progression of multiple sclerosis (MS), in which excessive activation of ferritinophagy was correlated to demyelination [12,131] and a persistent inflammatory state [12]. On the contrary, the possible involvement of ferritinophagy in other neurodegenerative conditions may be only speculative. Indeed, even if excessive iron levels have been detected in Alzheimer’s and Parkinson’s diseases [132,133], there is no direct evidence of a dysregulation NCOA4-mediated ferritinophagy. Additionally, the role of ferritinophagy in cancer remains elusive. Indeed, reduced expression of NCOA4 was found to be related to poor prognosis and cancer progression [134]. At the same time, investigations demonstrate that ferritinophagy contributes to tumor growth and metastasis [135].

## 7. Autophagy and Its Dual Role in the Regulation of Cancer

Autophagy plays a dual role in the regulation of cancer. Indeed, it can both promote the development of transformed cells and suppress tumorigenesis by limiting their survival and initiating cell death. Initially, autophagy was thought of as a tumor suppressor in normal cells, preventing the accumulation of radicles, controlling tissue damage, constraining inflammation, scavenging damaged organelles and macromolecules and thus reducing cancer cell metastases. However, increasing evidence supports the activation of oncogenes and the inactivation of tumor suppressor genes causes a reprogramming of the autophagy process in cancer. Moreover, autophagy also represents a normal cellular biological mechanism that benefits cancer cells and tumor development. Autophagy promotes the increased supply of nutrients to cancers cells, resulting in enhanced cancer cell survival. The following sections are dedicated to clarifying the controversial mechanisms of autophagy in cancer and its dual role in cancer development.

### 7.1. Tumor Suppression

In many tumor types, autophagy is downregulated, and therefore was initially considered a tumor suppressor. Common oncogenic alterations that support this position are the amplifications, loss of silencing, or gain-of-function mutations in Phosphatase and Tensin Homolog deleted on Chromosome 10 (PTEN), Phosphoinositide 3-kinases (PI3K) or AKT (also known as protein kinase B) [136], which function to inhibit autophagy by activating mTOR. In addition, arguably the most influential main tumor suppressor gene in our cells, p53, contributes to tumor growth through modulation of autophagy. Furthermore, p53 seems to have several functions in autophagy due to its functional localization. Cytoplasmic p53 can inhibit autophagy through protein–protein interactions with autophagic machinery [137]. Furthermore, nuclear p53 can induce autophagy through the transcription of autophagic machinery to modulate tumor suppression [86]. According to genomic studies of Beclin-11, ATG5 and ATG7, autophagy is directly involved in the suppression of spontaneous tumorigenesis. Indeed, their mutations in Beclin-11, ATG5 and ATG7 promotes tumor initiation [138,139,140]. Most notably, Beclin-1, which is a fundamental gene involved in the autophagy pathway, is also an important tumor suppressor. In breast cancer, Beclin-1 cooperates with a member of PI3KC3 complex, UVRAG, to recruit E-cadherin and other components of the E-cadherin/catenin complex to the membrane of breast cancer cells [38]. This process stimulates cell adhesion and evades the transcription of β-catenin-pathway genes, particularly tumor-promoting genes [141]. In human synovial sarcoma cells, the overexpression of Beclin-1 is associated with low cell viability, reduced cell proliferation and initiation of apoptosis [142]. Recent studies demonstrated that Beclin-1 is triggered by miR-140-3p in gastric cancer. Specifically, mir-140-3p inhibits cell survival, invasion and epithelial–mesenchymal transition (EMT) by controlling Beclin-1-associated autophagy [143]. Bae et al. analyzed the functional role of Beclin-1 in autophagy and demonstrated that in Hela cell lines, the reduction of the tumor protein, translationally controlled 1 (TPT1/TCTP), promoted autophagy by enhancing the cooperation between Beclin-1 and UVRAG, which stimulates autophagosome formation and maturation [144]. However, in human colon cancers, UVRAG monoallelic deletion is frequently observed, and its overexpression reduces cancer cell proliferation and controls tumorigenicity [38]. In addition to Beclin-1, several other autophagy genes were found to be critical in controlling tumorigenesis. For example, mice with biallelic deletion of Bif-1 (also known as SH3GLB1), which is involved in vesicle formation and membrane dynamics, developed spontaneous tumors and lymphomas [145]. Furthermore, the expression of ATGs, including ATG5 and ATG12, is associated with the inhibition of metastasis in breast cancer through the degradation of the neighbor to BRCA1 (NBR1) protein [146]. This suggests autophagy plays an important role in repressing tumorigenesis during the early stages of cancer development. The maintenance of genomic stability is critical for preventing tumor development. Moreover, autophagy is crucial in repairing DNA damage through nucleotide excision repair (NER), double stranded break repair (DDR) and controlling the activation of p38 to maintain genomic integrity [147,148,149]. It has been recently demonstrated that Glycyrrhizic Acid (GA), a natural triterpene, improves genomic stability by stimulating autophagy in UV-B irradiated human primary fibroblast cell lines (HDFs) [150]. Additionally, the control of Reactive Oxygen Species (ROS) is significantly involved in the interplay between autophagy and tumor suppression. Excess ROS are responsible for increased DNA damage and impaired mitochondrial function and facilitate the recruitment of tumor-promoting factors by altering the redox status of cells, thereby stimulating tumor development. However, higher basal levels of ROS in tumor cells decrease their resistance to excessive oxidative stress, while extremely high ROS levels are fatal to both normal and cancer cells [151]. Several chemotherapies stimulate both apoptosis in cancer cells and autophagy through ROS activation [152]. In addition, in vivo experiments show that autophagy loss can trigger extrinsic cell signaling of tumor cells, including facilitating the pro-tumorigenic inflammatory microenvironment [153,154]. Autophagy is regulated by different pathways, including the mTOR pathway. Studies mainly focus on traditional medicine compounds because of their anticancer properties, low toxicity and minimal side effects. Notably, eldecalcitol, a vitamin D analog, induces autophagy and apoptosis in human osteosarcoma MG-63 cells by increasing ROS production to inhibit the PI3K/Akt/mTOR signaling pathway [155]. In addition, daphnetin, a natural compound used for coronary heart diseases, could be used for ovarian cancer therapy, as it triggers ROS-induced cell death and activates autophagy by regulating the AMPK/Akt/mTOR pathway [156]. Moreover, in mouse models, the overexpression of selective autophagy receptor p62 (SQSTM1) upon autophagy loss promotes increased oxidative stress and tumor growth [157,158,159]. These data suggest that tumor suppression may play a role in several distinct autophagy proteins that function in different stages in the pathway, thereby underscoring that both the loss of autophagy is a hallmark of cancer, and that autophagy functions as a tumor suppressor.

### 7.2. Tumor Promotion

Studies have shown that autophagy enhances the stress tolerance of cancer cells after exposure to various conditions (radiotherapies, chemotherapies, hypothermic and hypoxic conditions), thus promoting their survival. Interestingly, increased basal levels of autophagy were detected in different cancer cell lines. Similarly, downregulating autophagy helps to reduce cancer cell survival. However, when referring to autophagy as a pro-tumorigenic mechanism, it is essential to specify that autophagy activation is highly dependent on the tumor type, the tumor stage and the genetic background of the individual with cancer [21]. Given that cancer cells have an increased metabolic demand caused by their excessive proliferation rate, autophagy enables their survival in unfavorable conditions by modulating tumor metabolism. Additionally, decreased activity of mitochondrial metabolism is associated with autophagy activation, increased proliferation and cellular migration in renal cell carcinoma [160]. Downregulation of Beclin-1 is correlated to the incremental reprogramming in glucose metabolism, cell invasion and proliferation in gastric cancer. Specifically, reduced Beclin-1 expression increases lymph node metastasis and is associated with poor prognosis [161]. When comparing tissue samples from Colorectal Cancers (CRCs) to normal colon tissue, Beclin-1 was downregulated in the CRC tissue, which promoted CRC cell invasion [162]. Conversely, Holah S. et al. demonstrated that Beclin-1 was upregulated in prostatic cancer and is associated with poor prognostic factors [163]. Other ATGs including ATG5, ATG7 and FIP200 were linked to human cancers [164,165,166]. Furthermore, recent evidence revealed that six polymorphisms ATG2B rs3759601, ATG16L1 rs2241880, ATG10 rs1864183, NOD2 rs2066844 and rs2066845 and ATG5 rs2245214 were associated with a predisposition to tumor development, particularly in glioblastoma [167]. In mouse models of KRAS-driven glioblastoma, stable knockdowns of ATG7, ATG13 and ULK1 demonstrated that autophagy is crucial for the initiation and sustained growth of glioma [168]. In breast cancer, autophagy promotes tumorigenesis. In mice, ATG17/FIP200 deficiency inhibited tumor growth [169]. In addition, the development, initiation and proliferation of melanoma was dampened when ATG7 was inactivated [159,170]. Overall, these findings suggest that the genomic integrity of ATGs is fundamental in preventing tissue damage, inflammation and cancer. Oncogenes promote cancer by enhancing mutations, enabling chromosomal translocation and inducing cell immortality. Neurotrophin-4 (NTF4) is upregulated in CRC and leads to poor overall survival. Silencing NTF4 in CRC cells decreased EMT by activating autophagy through the cooperation of ATG5 and the MAPK pathway [171]. Nuclear casein kinase and cyclin-dependent kinase substrates (NUCKS) are members of the high mobility group (HMG) family, which engage oncogenic properties in gastric cancer, and NUCKS are overexpressed in patients with poor prognosis. NUCKS silencing increases autophagy through the mTOR- Beclin-1 pathway, implicating NUCKS as potential therapeutic target in gastric cancer [172]. During cancer growth, autophagy stimulates cell survival and invasion, which leads to enhanced metastasis. In hepatocellular carcinoma (HCC) cell lines, the expression of MMP2 and MMP9 (enzymes that catalyze collagen hydrolysis) are upregulated after autophagy activation, which stimulates cell invasion [173]. Kyung Seok et al. reported, for the first time, how oxidized phospholipids influence tumor metastasis. As the main components of cell membrane, phospholipids are readily oxidized, which leads to increased cellular inflammation, modified cellular metabolism and increased intracellular stress [174]. Hepatoma and breast cancer cells with high levels of Oxidized Phospholipids (oxPLs) showed increased activation of autophagy, which improved the formation of metastatic lesions by promoting EMT [175]. In recent years, autophagy has been generally associated with the regulation of chemotherapeutic efficacy. Cisplatin, one of the main cancer drugs, has been demonstrated extensively to cause chemoresistance. In gastric cancer cell lines, the activation of autophagy by cisplatin reduced chemo-sensitivity [176]. Cancer consists of different populations of cells with dynamic metabolic reprogramming induced by the conditions of the microenvironment [177]. Markedly, tumor growth is characterized by high rates of glucose utilization. However, Wang et al. showed that although more glioblastoma cells are killed by chemotherapy after glucose starvation, a subpopulation of these cells increased autophagy to initiate quiescence, thus achieving survival and chemoresistance [178]. p62 overexpression and accumulation is a hallmark of impaired autophagy. The altered expression of p62 in tumor cells undergoing metabolic stress increases levels of ROS. Consequentially, these tumor cells with impaired autophagy can display compromised physiological function through DNA damage and genetic instability, enabling tumor progression. By suppressing autophagy, tumor cells did not enter quiescence and sensitivity to chemotherapeutic drugs was restored [178]. Therefore, autophagy is not only considered a mechanism of tumor cell survival, but also a promising therapeutic target for the treatment of different cancers. The following section highlights the role of autophagy in cancer after ECM detachment, and how activation of autophagy allows tumor cells to escape anoikis.

## 8. Autophagy in Specific Tumor Types

In the last several decades, we have made significant improvements in understanding the relationships between autophagy and almost all human cancer types. These studies have uncovered alternative therapeutic approaches and unveiled new potential biomarkers for early detection of several cancers. The following section summarizes the current understanding of autophagy in some cancer types with markedly high incidence and/or lethality.

### 8.1. Breast Cancer

Breast cancer can be classified into at least three subtypes that differ in incidence and progression, sites of metastasis and response to treatment. The majority of patients are breast cancer positive for estrogen and progesterone receptors (ER+ and PR+). This luminal tumor responds well to hormonal therapy and is associated with improved patient survival. The human epidermal growth factor receptor (HER)-2 enriched breast cancer is characterized by the overexpression of the ERBB2 oncogene, which leads to an aggressive tumoral phenotype. It is typically treated with anti-HER2 drugs including Trastuzumab, Pertuzumab and Lapatinib. Triple-negative breast cancer (TNBC) lacks the expression of hormone receptors, and HER-2 is an aggressive tumor which is highly susceptible to metastasis, chemoresistance, and is insensitive to the classical anti-breast-cancer therapies and radiotherapy [179]. An approved treatment for TNBC is the combination of nanoparticle of antitumor paclitaxel with the immunocheckpoint inhibitor, Atezolizumab [179]. Nevertheless, TNBC is characterized by high rates of metastases, primarily due to a small subpopulation of breast cancer stem cells which are particularly resistant to anticancer therapies, as differentiation sustains the tumor growth and induces tumor recurrence. Many reports demonstrate that autophagy plays an important role in breast cancer initiation, progression and therapy. Beclin-1 is deleted in more of 50–70% of sporadic human breast cancers, underscoring the involvement of autophagy in breast cancer. However, it is essential to specify that the Beclin-1 gene is on a chromosomal region that is frequently deleted in approximately 50% of sporadic breast carcinomas. This region also encodes for the tumor suppressor BRCA1. BRCA1 mutations dramatically increase the risk of developing breast cancer [180], which suggests the loss of Beclin-1 is not likely to be the primary cause of breast cancer. Beclin-1 overexpression in breast cancer cells reduced cell proliferation and tumorigenesis [181]. This supports the importance of Beclin-1 (and autophagy) in breast cancer development. In addition, it has been demonstrated extensively that Beclin-1 interacts with Bcl-2 in regulating the autophagy process. Interestingly, Bcl-2 is overexpressed in breast cancer [182] and its pro-survival and pro-tumorigenic functions in breast cancer are correlated to its capacity to inhibit autophagy by binding to Beclin-1 [183]. Further evidence of the importance of autophagy in breast cancer is demonstrated through studies of the essential autophagy protein FIP200. Its conditional knockout in a breast cancer mouse model increased the survival rate, and reduced tumor initiation, progression and metastasis. In this study, no increased apoptotic effects were observed; however, the FIP200 conditional knockout tumors had a reduced glycolytic microenvironment and increased expression of immune response genes, thereby suggesting an important role of autophagy in resolving the unfavorable tumor microenvironment present in breast cancer [169]. In addition, the autophagy protein ATG4A was identified as a mediator of breast cancer. By conducting a mammosphere formation RNAi screening, it was revealed that ATG4A was essential to maintain a sub-population of cancer stem cells, and to regulate the tumorigenic potential of breast cancer cells in vivo [184]. Interestingly, it has been highlighted that breast cancer stem cells utilize autophagy for surviving in the tumor microenvironment [185]. Indeed, Cufi and coworkers demonstrated that the knockdown of autophagy genes leads to a reduction of the number of cells with CD44(+) CD24(-/low) expression and, on the other hand, an increase in the expression of CD24 gene [186]. Furthermore, both in vivo and in vitro experiments remarked that during serum/nutrient deprivation condition, mesenchymal stem cells (MSC) promote the survival of breast cancer cells inhibiting the apoptosis [187]. In breast cancer, autophagy also contributes to the resistance of conventional therapies. Furthermore, defective autophagy sensitizes human mammary tumor cells to ER and oxidative stress, thus enhancing the sensitivity to anticancer therapy [188]. Resistance to therapy is also demonstrated by the high level of autophagy present in adriamycin-resistant MCF-7 cells [189]. Additionally, tamoxifen, the “gold standard” estrogen receptor modulator administrated to ER+ breast cancer patients, induces autophagy-mediated drug resistance [190,191].

HER2-therapy induces autophagy, promoting drug resistance as trastuzumab-resistant HER2+ breast cancer cells showed up-regulation of autophagy-related gene expression compared to the sensitive HER2+ cancer cells [192,193]. The sensitivity was restored by inhibiting autophagy in trastuzumab-resistant and lapatinib-resistant breast cancer cells, respectively [193,194]. Conversely, other reports showed that anticancer therapy activates autophagy to kill tumor growth. This has been demonstrated by treating human breast cancer cells with tetrandrine [195] or the known autophagy inducer Rottlerin, which promotes the cell death of breast cancer stem cells by mediating the suppression of Akt and mTOR signaling, and the up-regulation of phospho-AMPK [196].

### 8.2. Lung Cancer

Histologically, lung cancer is divided in two groups: small-cell lung cancer (SCLC, which accounts for about 15% of the total lung cases) and non-small-cell lung cancer (NSCLC), responsible for the remaining 85% of cases. Additionally, NSCLC is further subdivided into adenocarcinoma, squamous-cell carcinoma and large-cell carcinoma, which not only have different histological phenotypes, but also differ in the sites of origin and frequency. Adenocarcinoma is the most prevalent (about 50–70%) and is identified on the outer area of the lung. Large-cell carcinoma is found throughout the lung and accounts for 10% of cases. Squamous-cell carcinoma comprises about 20–30% of NSCLCs and usually occurs in the center of the lung close to the air tube.

Altogether, these lung cancer subtypes are the most prevalent cancer conditions among humans and represent the leading cause of cancer-related deaths worldwide. It was estimated that in 2018, there were about 2.1 million new cases and 1.8 million deaths. Primary causes of the high mortality of lung cancer are: (i) a poor prognosis due to late diagnosis resulting from a lack of clear symptoms, and (ii) high resistance to chemotherapy and radiation therapies. Therefore, it is essential to identify which molecular pathways are involved during the onset and progression of lung cancers so we may develop new diagnostic tools for early detection, and new therapeutic approaches. Interestingly, by sequencing the genomes of NSCLC patients, it was found that genes associated with autophagy and the regulation of mTOR activity (such as p53, PTEN, LKB1) are frequently altered or mutated in both adenocarcinomas and squamous-cell carcinomas, thus revealing an important role for autophagy in lung cancer [197]. This hypothesis was confirmed by adult mice knocked-out of essential autophagy genes that demonstrated decreased proliferation of cancer cells and increased tumor cell death of lung adenocarcinoma [198]. Consistently, human lung adenocarcinoma specimens showed increased autophagy compared to the matched normal lung tissues [199].

Another factor associated with poor clinical outcomes in NSLCLC patients is the paired-like homeodomain transcription factor 2 (PITX2) [200]. Silencing of PITX2B induces autophagy in lung cancer cells and simultaneously activates apoptosis to reduce tumor cell proliferation. Consistently, these effects were reversed by inhibiting autophagy with lysosomal inhibitors [201]. These findings reveal new insights into the role of PITX2B in lung cancer, and could unveil new therapeutic approaches based on the modulation of autophagy in those tumors characterized by high expression of PITX2B. Accumulating evidence indicated that up-regulation or down-regulation of microRNAs (miRs) may contribute to drug resistance in lung cancer cells [202]. A novel study demonstrates that the different levels of expression of miRNAs can regulate the sensitivity of lung cancer cells to conventional therapy by controlling autophagy. Indeed, when lung cancer cells were transfected with miR-101-3p, cisplatin resistance was reduced and autophagy was repressed, whereas the inhibition of miR-101-3p activates autophagy and reduces cisplatin sensitivity of lung cancer cells [203]. Moreover, autophagy also influences radiation sensitivity and radiotherapy efficacy. γ-irradiation in NSCLC cells causes histone protein modifications, including of H4K20me3, which was critical to target and increase the expression of several autophagy genes to protect lung cancer cells from radiotherapy [204].

Finally, the increased levels of autophagy in lung cancer cells were found to be highly correlated to the tumor microenvironment composed of stromal compartments (including non-tumor cells and fibroblasts). For example, a study that mimicked the tumor microenvironment in lung cancer cells showed that the proinflammatory cytokines IL-6 and IL-8 produced in the tumor microenvironment are strong involved autophagy inducers, which supports tumor growth. Indeed, IL-6 or IL-8 increased ATG5/12 conjugation and decreased p62, indicating cytokine-induced autophagy and, most importantly, autophagy inhibition suppressed tumor growth [205].

### 8.3. Mesothelioma

Malignant pleural mesothelioma (MM) is an aggressive malignancy of the serosal cavities. There are currently no effective therapies or early diagnostic strategies. The median survival of patients with MM varies from 8–15 months, and it is estimated that the global incidence is increasing and expected to reach its peak in this decade. From a histological point of view, MM is classified in three main sub-types: epithelioid (characterized by epithelioid-shaped cells), sarcomatoid (evidenced by spindle-shaped cells) and biphasic, which is characterized by a combination of epithelioid- and sarcomatoid-shaped cells in varying proportions [206]. From a clinical perspective, the epithelioid variant has the most favorable prognosis, while the sarcomatoid type is associated with poorer survival outcomes. The lack of effective early MM detection leads to the majority of patients suffering from advanced tumors at the point of their diagnosis when removal surgeries are more difficult and have increased risk. Therefore, chemotherapy (usually cisplatin combined with pemetrexed) is the first line of treatment for patients with MM, as radiotherapy is only effective in a low number of patients. In recent years, these interventions have been accompanied by lung-sparing surgery (extended pleurectomy and decortication) or extrapleural pneumonectomy [207]. However, despite these intervention strategies to decrease the morbidity and improve the clinical outcome the five-year survival rate following radical surgery remains notably low. The development of new therapeutic approaches and early diagnostics of MM is urgent, and therefore improving the understanding of the biological mechanisms of MM is essential. Several research groups have contributed to discovering the factors causing MM thus far, namely chronic exposure to asbestos, oncogenic viral infections and mutations of tumor suppressor genes. Autophagy plays a role in all of these factors contributing to MM. Truncating mutations in the tumor suppressor gene BRCA-1 associated protein-1 (BAP1) increased susceptibility to developing MM. Recent findings demonstrate BAP1 mutations cause a downregulation in intracellular Ca^2+^ dynamics and in mitochondrial bioenergetic functions [208,209]. Considering autophagy is a regulator, and is regulated by these cellular processes [210,211], recent reports hypothesized the relationship between BAP1 and autophagy in regulating MM development. Several studies have addressed an etiological role for Simian virus 40 (SV40) in MM. Interestingly, SV40 can activate autophagy to protect cancer cells exposed to high-stress conditions [212]. Currently, the correlation between SV40 and autophagy in MM development is poorly understood. Asbestos deposits remain in place in the pleura upon inhalation and cause both a chronic inflammatory response and autophagy activation to promote mesothelial cell transformation. Asbestos fibers are remarkably biopersistent and cannot be properly phagocytosed. As a result, autophagy activation and necrotic cell survival and proliferation promotes an inflammatory tumor microenvironment which enhances the production of pro-inflammatory mediators, ROS and genotoxic damage. Altogether, these factors eventually lead to the accumulation of mesothelial cells that undergo malignant transformation and form the primary tumor site where mesothelioma originates. Thus, the mesothelioma tumor microenvironment is a complex structure characterized by an intricate interaction between tumor, immune and stromal cells, making this disease very arduous to treat.

Among the diverse pro-inflammatory mediators, the protein high-mobility group box 1 (HMGB1) has become a pivotal concentration of study in MM. HMGB1 is a non-histone chromatin-binding protein mainly kept in the nucleus, but upon signals of cell damage, HMGB1 is released from the nucleus to the cytoplasm and the extracellular space where it functions to initiate and perpetuate the inflammatory response. It has been demonstrated that biopersistent asbestos fibers provoke the release of HMGB1 from primary human mesothelial cells, a fundamental event which drives mesothelioma growth and maintenance [213]. Evidence establishing the fundamental role of HMGB1 in driving MM onset and progression reports augmented levels of HMGB1 were detected in sera obtained from serum asbestos-exposed individuals and in MM patients [214,215]. Recent studies also demonstrated autophagy as an essential process during the malignant transformation which is induced by HMGB1 secretion and asbestos exposure. In addition, primary human mesothelial cells exposed to different asbestos fibers showed increased autophagy activation. High autophagy activation levels were correlated to the increased activity and secretion of HMGB1. Using mouse models characterized by reduced or knockout HMGB1, the fundamental importance of the HMGB1-autophagy axis in controlling the asbestos-induced malignant transformation of mesothelial cells was defined [215]. This work also showed autophagy machinery as a potential target to counteract malignant transformation of mesothelial cells exposed to asbestos. Moreover, pharmacological inhibition of autophagy significantly abrogated the HMGB1 signaling that controlled asbestos-mediated transformation [215]. Excessive autophagy activity was also detected in established MM [216,217]. These results revealed the possibility of using autophagy as a potential therapeutic target to overcome chemoresistance in MM. Recent works supported this hypothesis, demonstrating that by blocking autophagy with ULK1 antagonists or late-stage autophagy inhibitors, chemotherapeutic cytotoxicity was improved [218,219].

### 8.4. Pancreatic Cancer

Pancreatic cancers are highly aggressive tumors with an incidence of more 40,000 cases diagnosed per year. Due to the lack of effective therapies, the five-year overall survival of patients affected by pancreatic cancer is less than 5–9%. Late detection and diagnosis, as well as high therapeutic resistance to radiotherapy and chemotherapies, makes pancreatic cancer development and progression extremely difficult to manage. Pancreatic ductal adenocarcinoma (PDAC) represents 90% of all diagnosed pancreatic cancers. PDAC has a remarkably low mortality rate of 94% and is predicted to represent the primary cause of cancer-related deaths by 2030 [220].

In vivo and in vitro studies demonstrated that the autophagy process enhances the growth and survival of pancreatic cancer cells. Elevated levels of autophagy biomarkers were detected in pancreatic cancer cells lines and primary pancreatic tumors obtained from patients with PDAC. Moreover, these elevated levels of autophagy biomarkers were correlated with poor patient prognosis and survival outcome. The major role of autophagy in PDAC growth was further demonstrated through autophagy inhibition with specific molecules or genetic interference, as these methods suppressed cell proliferation in vitro and increased overall survival in mouse pancreatic cancer xenografts [221]. Nevertheless, the exact molecular mechanism modulating autophagy in pancreatic cancer still remains unclear. The YAP/TAZ signaling axis, which positively regulates autophagy by modulating the expression of the protein ARMUS [222], has been shown to increase the proliferation and differentiation of PDAC cancer stem cells [223]. Genetic mutations in KRAS are one of the primary causes of PDAC [224]. PDAC dependence on autophagy signaling is determined by mutated KRAS; suppressing KRAS may increase this dependency [225]. As a result, PDAC cells become acutely dependent on autophagy, an event that causes increased vulnerability for cancer cells and higher response to combined treatment with autophagy blockers and inhibitors of KRAS, or its effectors ERK and MAPK [225]. Aldehyde dehydrogenase (ALDH1) and osteopontin (OPN) were identified as two main contributors to controlling autophagy in PDAC [226]. PDAC is highly migratory and invasive, and the increased autophagy activity has been correlated to hypoxia-inducible factor-1α (HIF-1α) expression, which enhances pancreatic cancer metastases [227].

Although these in vivo and in vitro studies demonstrated that autophagy promotes pancreatic cancer cells to proliferate and survive and that its inhibition reduces tumor growth, clinical trials treating with autophagy inhibitors have not shown any improvement in PDAC management. This suggests it is essential to improve the mechanistic understanding of the role of autophagy in PDAC. Furthermore, we must not exclude that autophagy in PDAC, similar to other cancer types, may prevent tumorigenesis. Notably, autophagy protected pancreatic acinar cell functions from inflammation, ER stress and oxidative stress, which altogether contribute to PDAC development [228].

It is well known that the microenvironment contributes to the development of cancer [229]. Indeed, in PDAC, the autophagy can be induced by the hypoxic and nutrient-poor tumor microenvironment [230]. In particular, it was shown that the growth of PDAC is dependent on an interaction between pancreatic cancer cells and pancreatic stellate cells (PSCs), a specialized type of fibroblast that influence the progression of PDAC through a complex network of signaling molecules that involve extracellular matrix (ECM) proteins. In this case, PSCs activate autophagy to increase the production and secretion of metabolites to support the metabolism of pancreatic cancer cells [230].

## 9. Targeting Autophagy as Cancer Treatment

As previously described, autophagy is divided into three macroprocesses: initiation and nucleation, elongation and closure and vesicle degradation. Here, we outline the breakdown of the different known therapeutic modulators of autophagy and the role each of them plays during the key phases of the autophagic process (Table 1).

### 9.1. Initiation and Nucleation Phases

During the early stages of autophagy, tyrosine kinase inhibitors, mTOR inhibitors and BH3 agonists act as fundamental activators, whereas ULK1 and PI3K inhibitors interfere with this autophagy phase.

Tyrosine kinases (TKs) are enzymes that phosphorylate tyrosine residues of specific protein targets. These TKs regulate cell survival and proliferation in many different types of cancer. Therefore, the use of TK inhibitors interferes with cancer growth, and this interference has recently been linked to the modulation of the autophagy process [252]. For example, Erlotinib in NSCLC increases the levels of autophagy markers, such as LC3 and ATG5/7, by suppressing mTOR signaling and activation of p53, thus promoting cell death [231].

Sorafenib is the main chemotherapy used in hepatocellular carcinoma, and by modulating AKT activity, it controls molecular the switch from cytoprotective effects of autophagy to promoting autophagy-induced cell death (Figure 2) [232].

mTOR is a protein kinase involved in various cellular processes and, for this reason, it regulates various cellular mechanisms including autophagy, where it functions as the main repressor. Rapamycin, a metabolite isolated from *Streptomyces hygroscopicus* with antifungal and immunosuppressive activities, together with its analogues known as rapalogs (such as temsirolimus and everolimus), allosterically inhibit mTORC1 activity. Consequently, they influence autophagy activation and block tumor growth, which leads to cell death in different types of cancer [233]. However, due to their inability to inhibit the mTORC2 complex and other compensatory pathways that promote cell proliferation, their efficacy in managing tumor growth is limited (Figure 2) [234].

Other mTOR inhibitors are ATP competitors, which reduce the phosphorylation of mTOR target proteins and consequently block mTOR activity. An example is AZD8055, which can act as a substrate of both of the mTOR complexes and thus inhibits tumor proliferation [235].

BH3 mimetics represent a group of small molecules that mimic the interactions of BH3-only proteins. They can stimulate autophagy by interrupting the inhibitory action of Bcl-2 with Beclin-1. The mimetic Gossypol can induce both autophagy and apoptosis in malignant mesothelioma and colon cancer cells [253,254] by causing lethal mitochondrial damages throughout the opening of the mitochondrial permeability transition pore, a main regulator of mitochondrial apoptosis [255,256].

Obatoclax is another BH3-mimetic which interrupts MCL1 and Beclin-1 interaction, thereby provoking cell death via activation of necroptosis both in oral squamous cell carcinoma and acute lymphoblastic leukemia cells [236,237]. Furthermore, it has been demonstrated that obatoclax interferes with mTOR activities (Figure 2).

ULK1 is one of the main regulators of the autophagic process alongside AMPK and mTOR and is upregulated in various diseases, including cancer. Therefore, the inhibition of its activity has been studied as a possible antitumor therapeutic target, as reports show that ULK1 inhibition abrogates tumor growth and induces apoptosis. Different molecules that inhibit ULK1 activity have been developed, but most of them act as ATP-binding competitors. Notably, MRT68921 and MRT67307 strongly inhibit ULK1 and ULK2 in vitro and block autophagy in cells. ULK1 inhibition results in the accumulation of stalled early autophagosomal structures, which suggests ULK1 regulates the maturation of autophagosomes as well as initiation, and ULK1 may be a promising therapeutic target for an anticancer therapy [238]. In addition, SBI-0206965 inhibits ULK1 in various cancers, including renal cell carcinoma and neuroblastoma, and also blocks AMPK activity [238].

The family of PI3K is divided into three different classes depending on their substrates and different functions. Most reports primarily investigate class I, which inhibits autophagy through mTORC1 activation, and class III, which triggers autophagy [257]. The role of class II still remains unclear.

It is evident that most of these inhibitors do not act exclusively within a single stage of the autophagy pathway, and therefore their therapeutic effect cannot be attributed to the inhibition of autophagy alone.

The most significant therapeutic impact is observed from 3-methyladenine (3MA) treatment, and several studies report how 3MA improves the response to many chemotherapies. Its effect is twofold: in the presence of normal nutrition, it inhibits PI3KC1 and thus promotes autophagy activation; in starvation, 3MA constrains PI3KC3 and consequently, autophagy signaling will decrease [239]. 3MA must be used at high concentrations for the most effective functions, which is why its use in vivo is limited and its derivatives have been synthesized [240]. Another extensively used inhibitor is LY294002, which reduces autophagy and thus improves the cytotoxic effects of temozolomide treatment in melanoma cells (Figure 2) [241].

### 9.2. Elongation and Closure

The ATG family of proteins plays a fundamental role in the formation of the autophagosome vesicle. ATG inhibitors block autophagy by preventing the elongation and closure of autophagic vesicles. ATG is recruited via the PI3P membrane produced by VPS34 to enhance complex formation for elongation of the phagophore. For this reason, ATG inhibitors can target complex formation to lower autophagy levels in cancer.

NSC185058 binds to ATG4 and reduces the number and size of autophagosomes, and thus inhibits autophagy. Tumor growth was abrogated in a subcutaneous osteosarcoma model (Figure 2) [242].

Other inhibitors worth denoting are UAMC-2526, which promotes the efficacy of chemotherapy by inhibiting the autophagy process in murine colorectal cancer models [243], and tioconazole, an antifungal that acts on autophagy signaling by reducing cell viability and helping the antitumor effect of doxorubicin (Figure 2) [244].

### 9.3. Vesicle Degradation

To avoid the fusion of the lysosome with an autophagosome, or to block the activity of lysosomal hydrolases, lysosome inhibitors such as chloroquine and its derivates may be a promising therapeutic approach for cancer therapy. Drugs such as chloroquine (CQ) and hydroxychloroquine (HCQ) are already approved to treat various diseases, and only in recent years have they demonstrated potential as anticancer molecules [245]. As weak bases, their unprotonated forms can infiltrate organelles (in this case the lysosomes) by entering across the cell membrane; the increase in H+ concentration induces their protonation and increases the pH and the size of the lysosomes, thus blocking enzymatic functions. These inhibitors revealed an ability to improve chemotherapeutic effects and radiotherapy in several cancer types, both in vitro and in vivo. Notably, CQ and HCQ are the only two inhibitors that have entered clinical trials for cancer treatment, and their efficacy has been demonstrated in several studies. For example, CQ enhanced the cytotoxicity of chemotherapies in tumors where the PML-tumor suppressor is absent or downregulated [11]. CQ can overcome colon cancer chemotherapeutic resistance of bevacizumab and sensitize cells to death pathways [246]. HCQ decreases breast cancer cell survival when combined with tamoxifen or gefitinib, and plays an important role in melanoma cells, suppressing growth by synergizing with the mTOR inhibitor temsirolimus (Figure 2) [258].

Mefloquine (MQ) is another derivative that may be a potential cancer therapy. MQ is particularly effective in chronic myeloid leukemia, but also in prostate cancer and in some breast cancers. Bafilomycin A (BafA), which prevents the entry of H+ into the lysosomes by inhibiting the ATPase and blocking the autophagosome-lysosome fusion, decreased cell viability, increased invasive and migratory ability and promoted apoptosis of gastric cancer cells [247]. Furthermore, it is essential to emphasize the importance of the drugs already in use for other diseases to exert anticancer functions.

Although the molecular mechanism is still unclear, some antipsychotics or antidepressants seem to alter autophagic signaling by blocking the degradation of lysosomes and exhibit anticancer properties. For example, clozapine activates death in NSCLC [248]. Desmethylclomipramine, a metabolite of clomipramine, alters autophagic flux by blocking lysosomal degradation, which increases the sensitivity of cancer cells to chemotherapy [249]. Fluoxetine promotes the effect of paclitaxel in gastric cancer and inhibits cell proliferation in breast cancer (Figure 2) [250,251].

## 10. Conclusions

Autophagy is an intracellular immune response increasing cellular adaptation and maintenance of physiological homeostasis in response to environmental and cellular stresses. This process plays a dual role in cancer, suppressing it during the early stages and promoting it during the later stages. Indeed, the activity of autophagy seems be associated with some hallmarks of cancer, such as sustaining proliferative signaling, evading growth suppressors, resisting cell death mechanisms and activating invasion and metastasis. In particular, autophagy encourages cancer cells to escape anoikis and promote focal adhesion turnover, contributing to migration and dissemination of tumor cells in distant organs from the primary tumor. In addition, regulation of autophagy signaling at different stages can be used as a potential therapeutic approach for tumor prevention and therapy by abrogating tumor development and limiting its progression, thus increasing the efficiency of chemotherapy. Despite these factors, further investigations are essential to understand the possible consequences of autophagy manipulation in cancer therapy, due to the controversial role of autophagy in regulating cancer development.

## Figures and Tables

**Figure 1 biomedicines-10-01596-f001:**
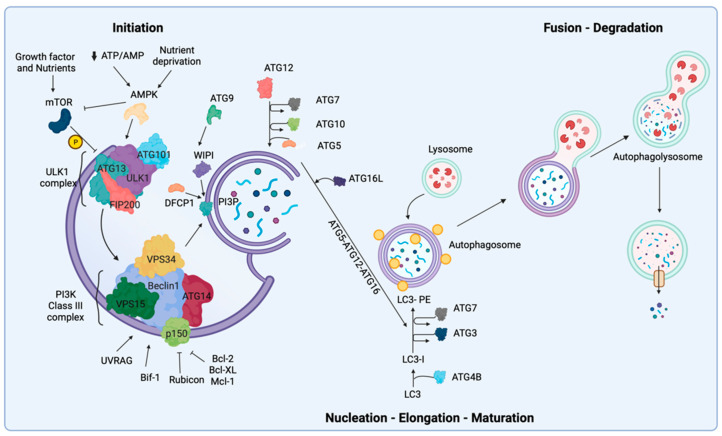
The regulation of mammalian autophagy. Under growth factors and nutrients, mTOR negatively regulates the initiation of autophagy by inhibiting the formation of the ULK1 complex (comprised of ATG13, ULK1, FIP200, and ATG101) through phosphorylation of the ATG13 subunit. Under nutrient deprivation, AMPK detects altered levels of the ATP/AMP ratio and downregulates mTOR activity, resulting in ULK1 complex activation. During phagophore nucleation, the ULK1 complex recruits and activates the PI3KC3 complex (which includes Beclin-1, ATG14, VPS 34, VPS15 and p150). Both negative regulators, Bcl-2, Bcl-XL and Mcl-1, and positive regulators, UVRAG, Bif-1 and Rubicon, can modulate PI3KC3 complex activity. The activation of PI3KC3 increases PI3P, which enhances PI3P interaction with WIPI and DFCP1. Elongation and maturation involve two ubiquitin-like conjugation systems. In the first system, ATG12–ATG5 binds and interacts with the ATG16L1 protein to form a multimeric complex: ATG5-ATG12-ATG16. The second system is orchestrated by LC3. LC3 undergoes a proteolytic cleavage by ATG4B proteases, which forms LC3-I. Consequently, via ATG7, ATG3 and ATG12-ATG5-ATG16, the PE are conjugated to LC3-I, which induces the formation of LC3-II and facilitates the closure of the phagophore. During fusion, the lysosomal membranes form autolysosomes by fusing with the autophagosomes. The contents of the autophagosome are then degraded by hydrolytic enzymes and the cellular components transform into building blocks that are transported from the lysosomal lumen into the cytosol.

**Figure 2 biomedicines-10-01596-f002:**
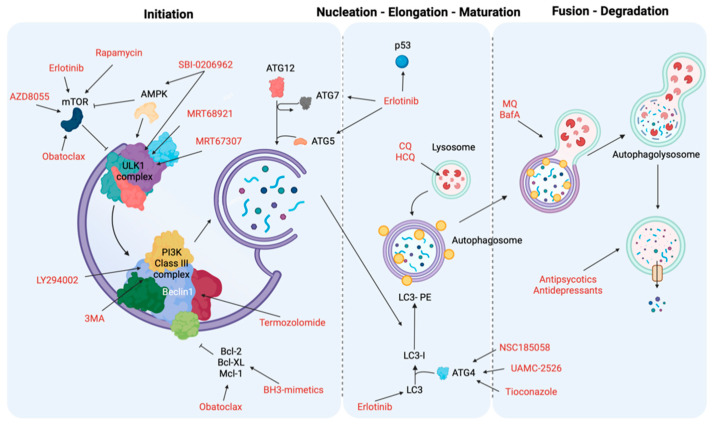
The modulators involved in the phases of autophagy: initiation and nucleation, elongation closure, and vesicle degradation. These factors can positively or negatively regulate autophagy by inhibiting cell growth and tumor progression and inducing of cancer cell death. For a detailed explanation, see the text.

**Table 1 biomedicines-10-01596-t001:** Targeting autophagy as a promising strategy for cancer treatment.

Autophagy Phases	Treatment	Target	Cancer Type	Ref.
*Initiation and nucleation phases*	Erlotinib	TK domain, mTOR and p53	NSCLC	[231]
	Sorafenib	AKT	Hepatocellular carcinoma (HCC)	[232]
	Rapamycin	mTORC1	Several	[233,234]
	AZD8055	mTOR	Several	[235]
	BH3 mimetics	BH3-proteins (Bcl-2, Mcl-1) and mTOR	Malignant mesothelioma, colon cancer cells, oral squamous cell carcinoma and acute lymphoblastic leukemia cells	[236,237]
	MRT68921 and MRT67307	ULK1 and ULK2	Several	[238]
	SBI-0206965	ULK1 and AMPK	Renal cell carcinoma and neuroblastoma	[238]
	3-methyladenine (3MA)	PI3K complex	Several	[239,240]
	LY294002	PI3K complex	Melanoma	[241]
*Elongation and closure*	NSC185058	ATG4	Subcutaneous osteosarcoma	[242]
	UAMC-2526	ATG proteins	Murine colorectal cancer	[243]
	Tioconazole	ATG proteins	Several	[244]
*Vesicle degradation*	Chloroquine (CQ) and hydroxychloroquine (HCQ)	Lysosomes	Colon cancer, melanoma cells and breast cancer	[245,246]
	Mefloquine (MQ) and Bafilomycin A (BafA)	Autophagosome-lysosome	Chronic myeloid leukemia, prostate cancer and breast cancers	[247]
	Antipsychotics or Antidepressants	Lysosomes	NSCLC, gastric cancer and breast cancer	[248,249,250,251]

## Data Availability

Not applicable.

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
