# Peer review of "Molecular Mechanisms of Autophagy in Cancer Development, Progression, and Therapy"

_biomedicines, 2022, doi:10.3390/biomedicines10071596_

Round 1

Reviewer 1 Report

In this review manuscript, authors summarized the current knowledges about autophagy mechanisms, described the role of autophagy activities in either promoting or suppressing cancer process, detailedly covered several specific cancer types that could be regulated by autophagy with different mechanisms, and finally explained the aspects that could be targeted for treating cancer through regulating autophagy. This review covered the many details regarding autophagy and cancer, but there are some points could be added to fully cover this specific topic.     

1. In terms of ‘Biological mechanisms of autophagy’, it would be better to include the transcriptional regulation of autophagy.

2. As most cancer types mentioned in this manuscript are solid tumors, so different roles of autophagy that played in the tumor microenvironment were worth more lines and paragraphs.

3. Topic about Anoikis seems singled out for some reason, while actually autophagy could be related with many different kinds of cell death, like apoptosis or ferroptosis.  

Author Response

Reviewer 1.

Comments and Suggestions for Authors

In this review manuscript, authors summarized the current knowledges about autophagy mechanisms, described the role of autophagy activities in either promoting or suppressing cancer process, detailedly covered several specific cancer types that could be regulated by autophagy with different mechanisms, and finally explained the aspects that could be targeted for treating cancer through regulating autophagy. This review covered the many details regarding autophagy and cancer, but there are some points could be added to fully cover this specific topic.     

  1. In terms of ‘Biological mechanisms of autophagy’, it would be better to include the transcriptional regulation of autophagy.

Author response: We thank the reviewer for the comments. We added the “transcriptional regulation of autophagy” for improve section of “Biological mechanisms of autophagy”. Furthermore, we reorganized the order of the different sections. Now, all the sections regarding the mechanisms that regulate autophagy have been moved before the cancer sections.

  1. As most cancer types mentioned in this manuscript are solid tumors, so different roles of autophagy that played in the tumor microenvironment were worth more lines and paragraphs.

Author response: We thank the reviewer for the observation. For each type of tumor, we added the roles of autophagy that played in the tumor microenvironment as requested.

  1. Topic about Anoikis seems singled out for some reason, while actually autophagy could be related with many different kinds of cell death, like apoptosis or ferroptosis. 

Author response: We thank the reviewer for the suggestions. Now, in this version of the manuscript, we added two paragraphs “Autophagy and apoptosis” and “Autophagy and ferroptosis” in correlation the autophagy to the other types of death. 

Reviewer 2 Report

Vitto et al., have submitted a review manuscript on autophagy in cancer. 

This is an interesting manuscript. I would appreciate the author's efforts in making such a nice piece of understanding of the molecular mechanisms of autophagy in cancer. 

Despite many review articles that have been published, still, there is room for discussing the novel molecular events happing in autophagy, particularly in cancer. 

Interestingly, the authors clearly delineated the role of autophagy in tumour development and tumour suppression, which budding oncologists need a lot. 

This reviewer doesn't have any major concerns, however, minor suggestions to improve the quality of the manuscript. 

- please correct "VSP34 and VSP15" in figure 1. They should be VPS34 & VPS15. Also in the main text.

- Line 244 "of of" delete one. 

I suggest the authors prepare a table summarizing the "Targeting autophagy as cancer treatment", despite the detailed text. This would attract more readers' attention. 

Author Response

Review 2

Comments and Suggestions for Authors

Vitto et al., have submitted a review manuscript on autophagy in cancer. 

This is an interesting manuscript. I would appreciate the author's efforts in making such a nice piece of understanding of the molecular mechanisms of autophagy in cancer. 

Despite many review articles that have been published, still, there is room for discussing the novel molecular events happing in autophagy, particularly in cancer. 

Interestingly, the authors clearly delineated the role of autophagy in tumour development and tumour suppression, which budding oncologists need a lot. 

This reviewer doesn't have any major concerns, however, minor suggestions to improve the quality of the manuscript. 

Author response: We thank the reviewer for having appreciates our work.

- please correct "VSP34 and VSP15" in figure 1. They should be VPS34 & VPS15. Also, in the main text.

Author response: We thank the reviewer for the observation. We have made the requested changes both in the text and in the figure. We also checked all acronyms throughout the manuscript

- Line 244 "of of" delete one.

Author response: We thank the reviewer for the suggestion. We have spelling mistakes.

I suggest the authors prepare a table summarizing the "Targeting autophagy as cancer treatment", despite the detailed text. This would attract more readers' attention. 

Author response: We thank the reviewer for the suggestion. We added a table for summarizing the "Targeting autophagy as cancer treatment”

Round 2

Reviewer 1 Report

The new version covered more aspects regarding the topic and organized well with more details. I recommend this version being accepted for publication.